# Design of an Electronic Nose System with Automatic End-Tidal Breath Gas Collection for Enhanced Breath Detection Performance

**DOI:** 10.3390/mi16040463

**Published:** 2025-04-14

**Authors:** Dongfu Xu, Pu Liu, Xiangming Meng, Yizhou Chen, Lei Du, Yan Zhang, Lixin Qiao, Wei Zhang, Jiale Kuang, Jingjing Liu

**Affiliations:** 1College of Automation Engineering, Northeast Electric Power University, Jilin 132012, China; 20162696@neepu.edu.cn (D.X.); 2202300635@neepu.edu.cn (P.L.); mxmforgo@gmail.com (X.M.); 2202300745@neepu.edu.cn (L.D.); 2202200607@neepu.edu.cn (Y.Z.); 2202300756@neepu.edu.cn (L.Q.); 2202300733@neepu.edu.cn (W.Z.); 2202300748@neepu.edu.cn (J.K.); 2Neuroscience Institute, Carnegie Mellon University, Pittsburgh, PA 15213, USA; yizhoucc@gmail.com

**Keywords:** electronic nose, end-tidal breath gases, exhaled breath analysis

## Abstract

End-tidal breath gases originate deep within the lungs, and their composition is an especially accurate reflection of the body’s metabolism and health status. Therefore, accurate collection of end-tidal breath gases is crucial to enhance electronic noses’ performance in breath detection. Regarding this issue, this study proposes a novel electronic nose system and employs a threshold control method based on exhaled gas flow characteristics to design a gas collection module. The module monitors real-time gas flow with a flow meter and integrates solenoid valves to regulate the gas path, enabling automatic collection of end-tidal breath gas. In this way, the design reduces dead space gas contamination and the impact of individual breathing pattern differences. The sensor array is designed to detect the collected gas, and the response chamber is optimized to improve the detection stability. At the same time, the control module realizes automation of the experiment process, including control of the gas path state, signal transmission, and data storage. Finally, the system is used for breath detection. We employ classical machine learning algorithms to classify breath samples from different health conditions with a classification accuracy of more than 90%, which is better than the accuracy achieved in other studies of this type. This is due to the improved quality of the gas we extracted, demonstrating the superiority of our proposed electronic nose system.

## 1. Introduction

Exhaled breath analysis is a method used in medical research and clinical detection. Human exhaled breath contains diverse volatile organic compounds (VOCs) and other gas molecules [1]. These molecules contain important information about the human body. The identification of disease-specific biomarkers enables health monitoring and disease prediction [2,3]. The metabolite profile of the human body in a disease state is different from that in the normal state. For example, elevated levels of methane and hydrogen are observed in patients with gastrointestinal disorders [4], and increased carbon monoxide concentrations are found in patients with respiratory diseases [5]. Disease conditions frequently induce intricate interrelationships among exhaled gas components. A precise understanding of the molecular composition of exhaled gases could significantly enhance diagnostic capabilities. However, the inherent complexity of these interactions poses challenges for accurate measurement and quantification, underscoring the need for advanced analytical tools. The electronic nose, fitted with an array of gas sensors corresponding to the characteristics of the gas composition to be measured [6,7], offers a sophisticated tool for analyzing breath composition. By combining the ‘gas fingerprint’ information output from the sensors with pattern recognition algorithms, the electronic nose can identify [8,9], detect [10,11], and quantify [12,13] the exhaled gas. Furthermore, this approach compensates for the cross-sensitivity issue of a single sensor. The advancements in gas sensor technology and artificial intelligence have accelerated the evolution of electronic noses for exhaled breath tests. Their noninvasive detection methods offer rapid and convenient diagnosis [14,15,16,17,18].

Human exhaled gas consists of end-tidal breath gases and dead space gases. End-tidal breath gases have completed gas exchange with the blood in the alveoli and provide more clinically relevant information about the body’s physiological state [19]. On the other hand, dead space gases do not participate in gas exchange during each breath, and they are mainly present in the anatomic dead space area, i.e., from the nose and mouth to the terminal bronchioles. The dead space gas’s composition is similar to that of the ambient air [20]. Therefore, breath analysis experiments should mitigate the influence of dead space gases to ensure the accuracy of results. Most existing studies do not consider this factor. Although a limited number of approaches have been proposed to address this issue, they all have different limitations.

Hao et al. [21] employed an electronic nose to analyze VOCs in breathed gas for lung cancer detection. In their study, subjects exhaled into a Tedlar bag. Since no exogenous VOCs in the physiological dead space were removed, the concentration of the targeted gas components was significantly reduced. To reduce the effects of dead space gases, Castellanos and Capuano et al. [22,23] proposed that subjects should first exhale part of the gas into the surrounding air and then exhale the remaining gas into the Tedlar bag [24]. This approach relies on subjective experience to remove dead space gas, which results in low efficiency and large errors. Grabowska-Polanowska and Mieth et al. [25,26,27] proposed using a carbon dioxide (CO2) sensor to differentiate and exclude dead space gases. However, water vapor in human exhaled gas can significantly impair the CO2 sensor’s detection accuracy. Additionally, squeezing the Tedlar bag is the conventional method of introducing the gas into the sensor chamber during the detection procedure. Different squeezing methods and strengths typically result in the issue of uncontrollable airflow [28]. Some studies have managed to control airflow by redesigning the chamber structure and introducing devices such as vacuum pumps. These solutions are complex and require frequent adjustments. Therefore, there is a pressing need to improve the system structure design and control mechanisms of traditional electronic nose systems.

In this study, we proposed a novel electronic nose system. The system incorporates a gas collection path based on a threshold separation standard, combining solenoid valves with a gas flow meter to ensure precise collection of end-tidal breath gases. Additionally, by optimizing the layout and control mechanisms between modules, we achieved autonomous control of the experimental process, eliminating issues with unsteady airflow. To validate the system’s performance, we selected the corresponding sensors according to the physical condition of the subjects to form the sensor array and conducted a human health status classification experiment, using machine learning algorithms (including Linear Discriminant Analysis (LDA) [29,30], Support Vector Machine (SVM) [31,32], K Nearest Neighbor (KNN) [33,34], and Random Forest (RF) [35]) to process the data. The experiment produced improved accuracy compared to related experiments. The results further verified the advantages of this system.

## 2. Electronic Nose Device

A structural diagram of the proposed electronic nose system is shown in Figure 1. The gas collection module consists of a mouthpiece, flow meter, solenoid valve, gas container, activated carbon filter, and vacuum pump. This setup enables the collection of the exhaled gases and the cleaning of the gas path. The sensor array, a crucial part of the detection module, is made of sensors that are sensitive to the target components in exhaled gases. Our test subjects were all young adults, averaging around 23 years old, with no major health conditions. Therefore, in assessing health status, we primarily focused on common ailments such as colds, diarrhea, and coughs, which involve the lungs, airways, digestive system, metabolic function, and certain chronic diseases. Based on the correlation between specific exhaled gas components and these health conditions, we carefully selected appropriate sensors—all of which were metal oxide semiconductor sensors (Weisheng, Zhengzhou, China). Digestive system disorders can be identified through exhaled gases such as hydrogen, methane, and hydrogen sulfide. To detect these gases, we utilized sensors such as MQ-138, MQ-8, MQ-136, MP-35, and MQ-4B, which exhibit high sensitivity to these compounds. Similarly, lung and airway diseases can be monitored by analyzing exhaled gases such as carbon monoxide and propane. For this purpose, we employed MQ-5, MQ-7B, MQ-9, and MP-135 sensors, which are particularly responsive to these gases. Additionally, metabolic disorders can be tracked through exhaled biomarkers such as acetone and ethanol, for which we selected MQ-138, WSP2110, and MP-135 sensors due to their sensitivity to these substances. Details on the sensors and the gases they detect are shown in Table A1. The system control module manages the gas path states and data transmission. The control module consists of a relay, a solenoid valve, a flow meter, a data acquisition board, a power module, and host computer software.

Through systematic structural design, each module is optimally integrated to meet various functional requirements. The gas container is directly connected to the sensor chamber through a solenoid valve, and the process of the collected gas entering the chamber is completely isolated from the external environment. This design prevents the introduction of ambient VOCs, thereby maintaining the integrity of the sample. This real-time dynamic process also prevents condensation caused by the gas remaining in the gas container for too long.

### 2.1. Gas Collection Module Design and Methodology

During a full exhalation, the ratio of end-tidal breath gas to dead space gas is approximately 7:3 [1]. This relationship can be verified using the Bohr equation [36], which leverages the dilution effect of carbon dioxide partial pressure. Since dead space gas does not participate in gas exchange, it does not absorb carbon dioxide from the blood; end-tidal breath gas is the sole component actively involved in this process. By measuring the difference between the partial pressure of carbon dioxide in exhaled gas and that in arterial blood, the proportion of dead space gas within the full exhalation can be accurately determined. Through extensive experimentation and analysis, it has been determined that dead space gas typically constitutes around 30% of an individual’s total exhaled gas [37], while end-tidal breath gas accounts for 70%. This corresponds to an end-tidal breath gas to dead space gas ratio of approximately 7:3. This finding has also been corroborated by anatomical studies [38]. Rohrer et al. reported that the volume of anatomical dead space is approximately 150 mL, whereas the total exhaled gas volume is around 500 mL. Today, this conclusion is widely recognized and applied in the field of respiratory medicine [39]. Based on this feature, a complete gas collection path was designed using the ratio of dead space gas to end-tidal breath gas content as the separation criterion (Figure 2). Since spirometry varies between subjects, setting the same separation threshold for all would result in significant errors. To address this, a pre-experiment should be conducted for each subject to determine their corresponding threshold before gas collection.

In the gas collection path, a flow meter is installed between the mouthpiece and solenoid valve A. We utilize a high-precision MENS series gas flow meter with a measuring range of 0 to 500 L/min and a working pressure of below 1 MPa. The measurement accuracy is within ±1% FS, ensuring reliable performance. The flow meter features a compact design, with a height of 12 cm, a length of 9 cm, and a width of 3.5 cm, making it well suited for integration into the electronic nose system. It measures the exhaled gas flow and transmits the flow data to the host computer. The solenoid valve is a three-way valve of the FQY-23 series with polytetrafluoroethylene (PTFE) internal components. One end of the valve is connected to an activated carbon molecular sieve, which filters out interfering gases from the ambient air during the cleaning of the device, The other two ends are used to control the opening and closing of the gas line to the gas container. The internal connections and the gas container are also made of PTFE tubes. PTFE exhibits extremely high chemical inertness and does not adsorb volatile compounds from the sample gas. A vacuum pump is employed to generate a negative-pressure environment to clean the device.

The gas collection path has three states: a cleaning state, a dead space gas removal state, and an end-tidal breath gas collection state. During the cleaning state, external air enters the device from the port of the activated carbon molecular to clean the system. This process cleanses the system by flushing out residual gases and contaminants, preparing the device for accurate sampling. During the dead space gas removal state, the anterior gas exhaled by the human body enters the collection device from the left side and exits through the right side. This ensures that dead space gases are excluded from subsequent analysis. During the collection state, the subsequent exhaled gas enters from the left side and is stored in the gas container. The solenoid valve connects its left port to the right port when powered and connects its upper port to the right port when not powered.

The schematic diagram of gas distribution and flow change in the gas path in the end-tidal breath gas collection state is shown in Figure 2. At this stage, the gas flow value is lower than the threshold value, and the dead space gas has all been discharged through the gas path. The end-tidal breath gases are stored in the gas container, where it is compressed to achieve enrichment as the gas accumulates.

### 2.2. Sensor Chamber Design

The design of a stable and controlled chamber is critical for improving the accuracy of gas detection. The chamber is made of PTFE material, which is chemically stable and non-toxic and does not produce interfering gas molecules. To improve chamber tightness, a grooved design at the top cover is used to secure the fit between the cover and the chamber body. The size of the response space of the chamber is minimized to prevent the response space from being too large for adequate contact between the gas and the sensor. In addition, considering gas dynamics, all corners within the chamber are rounded to reduce dead space and facilitate the cleaning and detection processes. The electronic nose circuit board is placed inside the chamber, with the sensor array facing downwards, 3 mm away from the bottom.

### 2.3. System Control Flow

In the electronic nose control system, the host computer is the primary control unit, communicating with the actuating components (solenoid valve and vacuum pump) via a data acquisition board. The system operation is divided into two stages: the pre-experiment stage and the formal experiment stage (Figure 3). During the pre-experiment stage, the subject exhales a complete breath sample into the collection device. The host computer reads the data from the flow meter and calculates the threshold value; this process is repeated three times to take the average value as the final threshold value output. In the formal experiment stage, the host computer first activates the vacuum pump to clean the gas path and then switches the collection device to the dead space gas removal state. The subject begins exhalation, and when the gas flow drops below the preset threshold, the device automatically transitions to the end-tidal breath gas collection state. Once exhalation ends, the device switches to the detection state, directing the gas into the chamber. The sensor array then outputs response data, which is transmitted to the host computer for storage. This automated control process enhances experimental efficiency and minimizes human error.

A 10-min break is scheduled between the pre-experimental stage and the formal experimental stage to allow the subject’s lung gas concentration to return to a stable level. This rest period helps minimize bias caused by biomarker consumption, ensuring that the exhaled gas collected during the formal experiment is more representative and consistent.

To ensure that the gas enters the sensor chamber at a constant flow rate and reduces the impact of airflow fluctuations on sensor accuracy, a PWM-controlled vacuum pump is used in this setup. Pulse Width Modulation (PWM) is a technique that utilizes the digital output of a microprocessor to control analog circuits. It has two key adjustable parameters: frequency and duty cycle. The frequency defines how many times the signal transitions from high to low and back to high per second, while the duty cycle represents the proportion of time the signal remains at a high level within one cycle. At a fixed frequency, varying the duty cycle allows different analog output voltages. Consequently, most motors regulate their speed by adjusting the duty cycle—the higher the duty cycle, the faster the motor speed. In this experiment, we reduced the duty cycle of vacuum pump E when transitioning from the cleaning state to the detection state. During the cleaning phase, a higher duty cycle was used to enhance efficiency and cleaning quality. However, in the detection phase, the motor must create a negative pressure environment within the detection chamber to introduce the sample gas. If the gas flow rate is too high, the sample may not have sufficient contact with the sensor, affecting the diffusion and adsorption of gas molecules on the sensor’s sensitive layer, ultimately leading to detection errors. To mitigate this issue, we lowered the duty cycle during this stage. Once the experiment enters the detection state, the vacuum pump’s speed is decreased by lowering the duty cycle of the PWM wave.

### 2.4. Detection Module Circuit Design

The electronic nose circuit board is composed of two primary sections: the power supply circuit and the sensor peripheral circuit. As most sensors in the system require a 5 V heating voltage and a reference voltage, a 5 V switching power supply circuit (Figure 4c) and a reference power supply circuit (Figure 4b) are designed to meet these requirements. The 5 V switching power supply, utilizing a TPS56528 chip (YouXin Electronic, Shenzhen, China), operates with an input voltage range of 4.5–18 V, an output voltage range of 0.6–7 V, a switching frequency of 650 kHz, and an output current capacity of 5 A. The reference power supply uses a REF5050 chip (YouXin Electronic, Shenzhen, China) to provide the necessary operating voltage for the sensors. Additionally, for sensors such as the MP9 used in methane gas detection, which require a heating voltage of 1.5 V ± 0.1 V, an adjustable switching power supply circuit is also designed (Figure 4d). The adjustable switching power supply circuit employs a TPS562200 chip (YouXin Electronic, Shenzhen, China). According to the special operating temperature conditions of the sensor, it is heated one minute before detection. The sensor peripheral circuit is shown in Figure 4a, the response output from the short-circuited pins 4 and 6 is transmitted to the data acquisition board for digital signal conversion.

## 3. Experimental Operation

### 3.1. Experimental Design

We established strict experimental procedures and precautions for the classification of human health conditions. A self-assessed health rating scale was used for quantitative analysis [40], and the questions set in the scale are shown in Table A2. Each question is rated on a scale from 0 to 10, with a total of 11 levels. Participants assign scores based on their own condition, where a lower score indicates a more severe presence of the associated disease. The final score is calculated as the sum of the six individual question scores. Based on this total score, health status is classified into three levels: below 45 is considered “poor”, 46 to 55 is classified as “good”, and 56 or above is rated as “excellent”. Self-assessed health rating scale is a subjective evaluation of an individual’s health status, and researches have indicated that it is strongly correlated with the actual health status [41], which can reflect the health level of the organism [42]. In addition, human health conditions were classified using our designed electronic nose in conjunction with some machine learning algorithms [33].

To control confounding factors, the selected subjects were all 23 years old, had no major diseases, and had no unhealthy habits (smoking, drinking, etc.). In addition, relevant experimental steps were designed to avoid the influence of exercise, medication, disease, and diet on the exhalation spectrum. The experiment lasted for ten days, and a total of 115 sets of data were collected. All subjects were notified of the purpose of the experiment in advance and signed informed consent forms.

The complete experimental procedure consisted of five stages: determination of the separation threshold, gas path cleaning, gas collection, sample detection, and data processing. To minimize the impact of temperature, we preheated the sensor before detection using a designed power supply circuit, ensuring that the sensor reached its optimal operating temperature. To reduce the influence of humidity, we continuously purged the detection chamber with gas until it reaches the detection state, maintaining a ventilated environment to prevent water vapor condensation.

### 3.2. Determination of the Separation Threshold

Figure 5 presents a schematic diagram of the expiratory flow versus time curve during the pre-experiment stage. The threshold value for each subject can be determined using the threshold separation criterion. The vertical coordinate at the threshold point in the figure represents the required threshold value. The specific method is as follows:

Firstly, the total amount of exhaled gas can be calculated from the flow value data: (1)$$V_{1}+V_{2}=\int_{0}^{t}Qdt$$
where *Q* is the value of the gas flow, $V_{1}$ is the volume of end-tidal breath gas, and $V_{2}$ is the volume of dead space gas. Equation (Equation 2) is used to determine the volume of $V_{2}$, and Equation (Equation 3) is used to determine the time point $t_{1}$ at which all of the dead space gas has passed through the flow meter.(2)$$V_{2}=\frac{3}{10}\left(V_{1}+V_{2}\right)$$(3)$$\int_{0}^{t_{1}}Qdt=V_{2}$$

The distribution of gas in the collection device at the moment $t_{1}$ is shown in Figure 6. To accurately collect the end-tidal breath gas, it is necessary to combine the volume of the gas container to solve for the time point $t_{2}$ when all the dead space gas passes through solenoid valve B. The method is given in the following equation: (4)$$\int_{0}^{t_{2}}Qdt=\int_{0}^{t_{1}}Qdt+V_{3}$$
where $V_{3}$ is the volume of the gas container and the flow value at time $t_{2}$ is the separation threshold.

## 4. Experimental Results and Discussion

### 4.1. Data Feasibility Analysis and Preprocessing

To investigate the differences in the patterns of sensor data under different health conditions, this study selected the integral ratio of the output data from each sensor as the feature and plotted the box plot shown in Figure 7. When comparing the box plots of different sensors under the same health condition, clear differences between the sensors are observed. This result shows that each sensor responds differently to specific gas compositions. Focusing on the box plots of the same sensor under different health conditions, in terms of the measured data’s median, mean, and interquartile range, there are notable variations in the data distribution across different health conditions. These differences demonstrated that the electronic nose system can differentiate between distinct health statuses effectively.

The raw signal collected during the measurement is the analog voltage output from each gas sensor. Because changes in temperature and humidity in the environment may affect the performance of the sensor, this study uses the conductivity ratio instead of the raw sensor voltage. Additionally, this study employs multiple sensors in an array, each with varying original voltage outputs. By using the conductivity ratio as a standardized index, data from different sensors can be normalized onto the same scale, facilitating data fusion, subsequent processing, and machine learning modeling. The formula of conductivity ratio is shown in Equation (Equation 5).(5)$$R_{g}=G_{\mathrm{gas}}/G_{air}$$
where $G_{air}$ is the conductivity value in clean air and $G_{\mathrm{gas}}$ is the conductivity value in the measured gas.

To effectively remove data noise and reduce redundant information, we performed the following preprocessing steps on the collected data. First, we employed mean filtering to reduce noise in the original data. Next, we extracted five features, including the integration ratio, the maximum location, the maximum first-order derivative, and two frequency-domain eigenvalues.

### 4.2. Classification and Model Validation

Electronic nose breath detection experiments frequently employ classical machine learning algorithms such as LDA, SVM, RF, and KNN, which perform exceptionally well when handling non-linear data. In this study, these traditional algorithms were employed to facilitate comparisons with similar experiments and validate the effectiveness of the electronic nose device in collecting end-tidal breath gases. Traditional machine learning algorithms are inefficient when dealing with high-dimensional data, and overfitting may occur. To solve this challenge, we combined dimension reduction method to process the data. First, the extracted five feature value data are processed using the LDA method for dimensionality reduction, reducing the original 50-dimensional features to 2-dimensional. The resulting 2-dimensional data are then fed into SVM, RF, and KNN models for classification [43]. The models’ performance was assessed using classification accuracy, recall, precision, and the F1 score. While accuracy reflects overall correctness, recall, precision, and F1 score provide more reliable evaluation in imbalanced health testing tasks by emphasizing missed detections, false positives, and the balance between them.

In addition to LDA for dimensionality reduction, the PCA method was also applied to reduce the original 50 dimensions of the feature data to 3 dimension. The reduced data were then fed into the classification model for validation. However, the classification results after PCA dimensionality reduction were not satisfactory due to the nonlinear nature of the electronic nose data. The data distribution after PCA dimensionality reduction is shown in Figure A1, and the corresponding classification results are presented in Table A3.

LDA projects each sample $x_{i}\in R^{50}$ of the 50-dimensional feature data to 2 dimensions, and finds the optimal projection basis vectors $\left(\omega_{1},\omega_{2}\right)$ that maximizes the separation between classes for projection to 2 dimensions. $S_{w}$ is the total within-class scatter matrix and $S_{b}$ is the between-class scatter matrix. The formula is as follows: (6)$$S_{w}=\sum_{j=1}^{3}\sum_{x_{i}\in C_{j}}\left(x_{i}-\mu_{j}\right){\left(x_{i}-\mu_{j}\right)}^{T}$$(7)$$S_{b}=\sum_{j=1}^{3}N_{j}\left(\mu_{j}-\mu \right){\left(\mu_{j}-\mu \right)}^{T}$$

In the above equation, $C_{j}$ is the sample of the *j*th class and $\mu_{j}$ is the mean vector of the sample of the *j*th class. To maximize the ratio of between-class scatter to within-class scatter, it is converted to the following equation: (8)$$J=\frac{\omega^{T}S_{b}\omega }{\omega^{T}S_{w}\omega }$$

This is transformed into the following feature-solving problem in the following equation: (9)$$S_{b}\omega =\lambda S_{w}\omega$$

The eigenvectors $\left(\omega_{1},\omega_{2}\right)$ corresponding to the first two largest eigenvalues are selected and the data are projected to obtain the following data output: (10)$$y_{i}={\left[\omega_{1}^{T}x_{i},\omega_{2}^{T}x_{i}\right]}^{T}\;\;\;\;\;i=1,2,....N$$

The distribution of the data after LDA dimensionality reduction is shown in Figure 8. The complexity of the data is reduced while the most discriminative features are extracted. The dimensionality-reduced data are then fed into the classification model for classification.

SVM seeks to identify the optimal hyperplane that divides different data classes by maximizing the margin between them [44]. We start with the given training samples: (11)$$T=\left\{\left(x_{1},y_{1}\right),\left(x_{2},y_{2}\right),...,\left(x_{i},y_{i}\right)\right\}$$
where $x_{i}$ is the *i*th feature vector and $y_{i}$ is the class marker; here, $x_{i}\in R^{2}$, i=1,2,3,...,N$y_{i}\in \left\{-1,+1\right\}$, i=1,2,3,...,N; when the class marker is equal to +1, it is a positive case, and when it is equal to −1, it is a negative case. We classify them using the following linear function: (12)$$f\left(x\right)=\omega^{T}x+b=\sum_{i=1}^{n}\omega_{i}x_{i}+b$$

The primary objective of the support vector machines is to minimize the objective function: (13)$$\min\frac{1}{2}{∥\omega ∥}^{2}+c\sum_{i=1}^{n}\zeta_{i}$$(14)$$s.a.y_{i}\left(\omega^{T}x_{i}+b\right)\geq 1-\zeta_{i},\;\;\;\;\;\;\;\;\;\;\;\zeta_{i}\geq 0$$

For the nonlinear problem in this experiment, the data are mapped to a higher dimensional space by introducing a kernel function to find a linearly divisible hyperplane in the higher dimensional space. The model’s decision function is(15)$$f\left(x\right)=\sum_{i=1}^{n}\alpha_{i}y_{i}K\left(x_{i},x\right)+b$$
where $K(x_{i},x)$ is the kernel function, the radial basis kernel function (RBF) is selected as the kernel function of the model after experimental validation with the specific formulae into the following: (16)$$K\left(x_{i},x_{j}\right)=\exp(-\gamma ||x_{i}-x_{j}\left|{|}^{2}\right)$$

Since this experiment is a multi-classification problem, a ‘one-vs.-one’ approach is adopted to establish a multi-classification model, $\frac{n(n-1)}{2}$ support vector machines are constructed, and each support vector machine is trained with two different categories of data. The final classification result is determined through a majority voting mechanism [45].

The performance of an SVM heavily relies on the hyperparameter settings, and if the *c* and γ parameters are not properly combined, the model may suffer from overfitting or underfitting. Therefore, this study uses Genetic Algorithm (GA) to determine the optimal hyperparameters of the SVM model. Each individual in the population represents a set of candidate parameters and its fitness value is calculated by a fitness function to measure the merit of this parameter combination. The genetic algorithm gradually finds the individual with the best fitness value through operations such as selection, crossover, and mutation. As shown in Figure 9, the SVM model reaches the highest performance after 7 generations of GA execution.

To verify the effectiveness of genetic algorithm optimization, we randomly selected five sets of parameter combinations within the same parameter range as the optimization algorithm as the parameters before optimization for comparative validation. The classification results are shown in Table 1.

KNN works on the principle of classification based on the similarity between sample points. Firstly the sample to be classified is fed into the model and we use Euclidean distance to calculate its distance to the sample points in the training set. The formula is as follows: (17)$$d\left(x^{\prime },x_{i}\right)=\sqrt{{\left(x_{1}^{\prime }-x_{i1}\right)}^{2}+{\left(x_{2}^{\prime }-x_{i2}\right)}^{2}}$$

A set containing the distances from all sample points in the training set to $x^{\prime }$ is generated: (18)$$D^{\prime }=\{\left(x_{1},d\left(x^{\prime },x_{1}\right)\right),\left(x_{2},d\left(x^{\prime },x_{2}\right)\right),...,\left(x_{n},d\left(x^{\prime },x_{n}\right)\right)$$

The K neighbors with the smallest distances are selected from the distance set $D^{\prime }$ to determine the class of the sample [46].

RF forms a forest by building multiple decision trees [47]. Different decision trees are independent to each other, and when a new sample is input, each decision tree in the forest is allowed to make a classification judgment separately. The final result is decided by ‘voting’ of all trees.

We used five-fold cross-validation to split the dataset into five subsets for training and testing. Table 2 presents the classification results of three algorithms on the test samples.

The experimental results show that the classification accuracy of all three algorithms is greater than 90%, indicating that our electronic nose system can effectively differentiate different health conditions in the human body. Besides, the recall of each category is also above 88%. The F1 score combines the precision and recall of the model, and the results of each category are higher than 89%, which effectively excludes the influence of category imbalance on the assessment of the model’s real performance. Many existing studies used electronic noses for breath testing, often using algorithms such as KNN, RF, LDA, and SVM to process the data [12,48,49,50,51]. They have shown successful results in differentiating between diseases such as diabetes, renal disorders, airway diseases, and cancer. However, the classification accuracies of these studies were between 70% and 90%. In contrast, we have achieved significantly better experimental results in this study. The reason lies in the optimal design of the collection module of the electronic nose system. By correlating the exhaled gas flow with the gas separation threshold and conducting a pre-experiment to account for inter-subject variations in lung capacity, we ensured a precise collection of end-tidal breath gases, which are more directly related to human health status. This method improves the quality and representativeness of the experimental data from the source, providing a strong foundation for accurate and reliable classification results.

## 5. Conclusions

This study designed an electronic nose system that can automatically collect end-tidal breath gases and predict the subject health condition from the collected gas with high accuracy. A threshold separation method based on expiratory flow characteristics was proposed. Based on this method combined with the solenoid valve to control the gas path state, accurate end-tidal breath gas collection was achieved. Structural and control method improvements further minimized external interference and allowed for full automation of the experimental process. Breath tests were performed on subjects with different health conditions and the data were processed using LDA-SVM, LDA-RF, and LDA-KNN classification algorithms. The results show that the classification accuracies of the test set are over 90%, outperforming similar studies. With the same experimental process and classification algorithms, the excellent performance of this study is mainly due to the enhancement of the raw data collection stage, where we successfully isolated the end-tidal breath gases that are more representative of the human health condition. Compared to other existing studies, we have a higher accuracy of end-tidal breath gas collection by the proposed threshold separation method.

In summary, the electronic nose system developed in this study shows strong potential for end-tidal breath gas collection and health status classification. Because the samples collected are of higher quality and the differences between individuals are more clearly expressed, it is easier for the model to learn the characteristic patterns that are truly associated with the target disease, rather than noise or confounding factors. Moreover, the signal intensity of VOCs in the end-tidal breath gas is more concentrated, and the eigenvalue data are more prominent, which further improves the performance of the model. To further enhance detection performance, future research will integrate pulse monitoring devices and heart rate testing equipment, utilizing data fusion techniques to provide more comprehensive and accurate monitoring of human health status. Additionally, deep learning methods will be introduced to handle larger and more diverse datasets resulting from the fusion process. Moreover, the system’s structural design will be further optimized by incorporating a microcontroller to enhance portability and adaptability.

## Figures and Tables

**Figure 1 micromachines-16-00463-f001:**
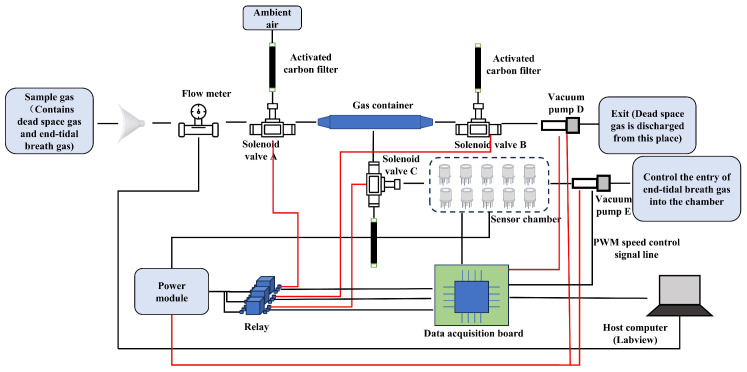
Structural diagram of electronic nose system.

**Figure 2 micromachines-16-00463-f002:**
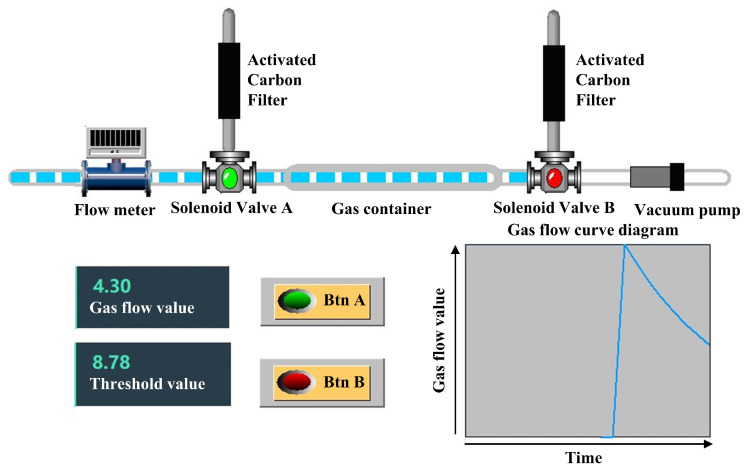
Schematic diagram of end-tidal breath gas collection state.

**Figure 3 micromachines-16-00463-f003:**
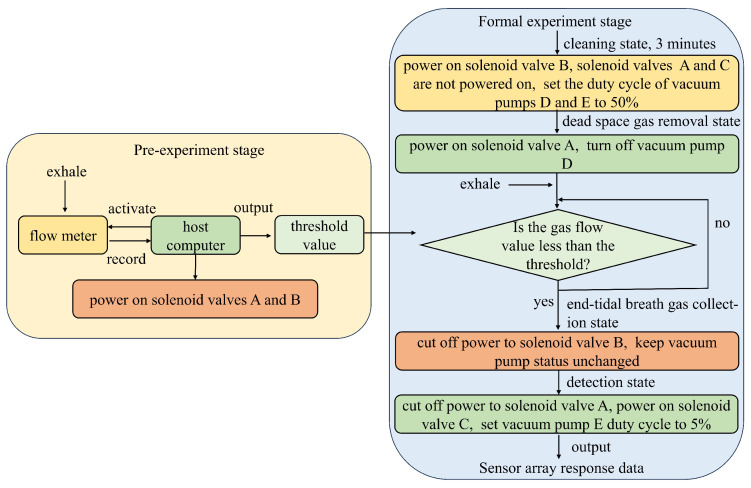
System control flow chart.

**Figure 4 micromachines-16-00463-f004:**
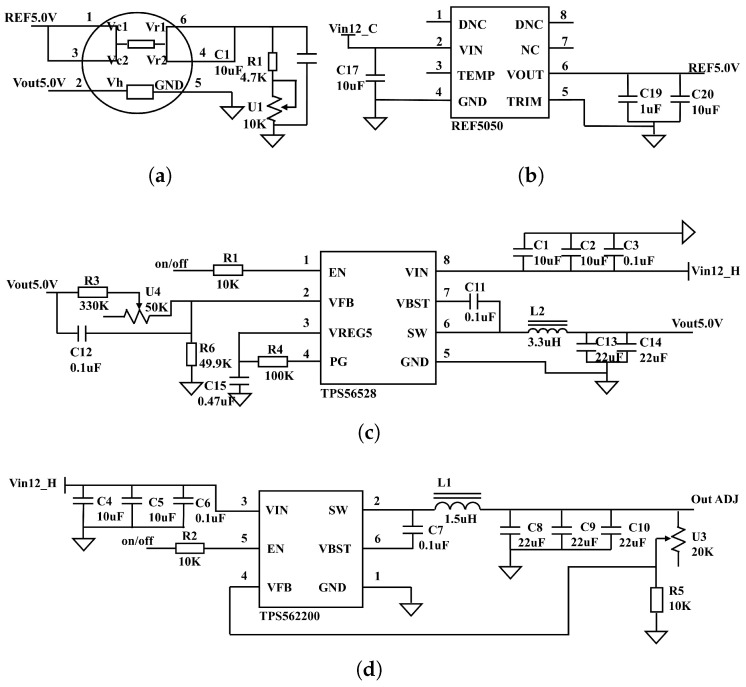
(**a**) Sensor peripheral circuit; (**b**) reference power supply circuit; (**c**) 5 V switching power supply circuit; (**d**) adjustable switching power supply circuit.

**Figure 5 micromachines-16-00463-f005:**
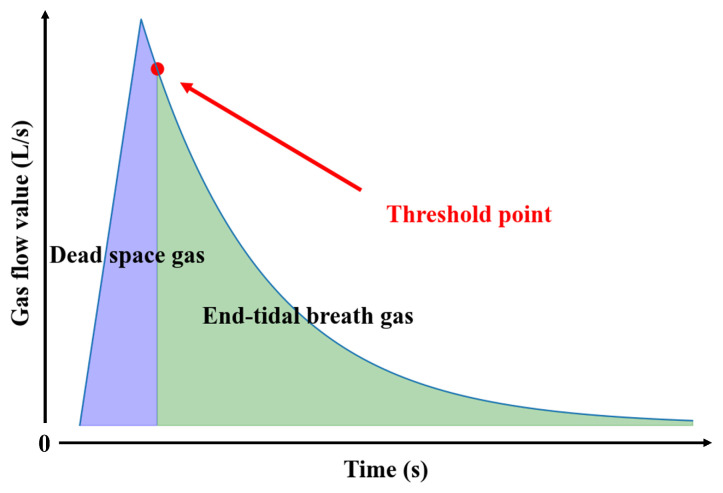
Schematic diagram of the expiratory flow–time curve.

**Figure 6 micromachines-16-00463-f006:**
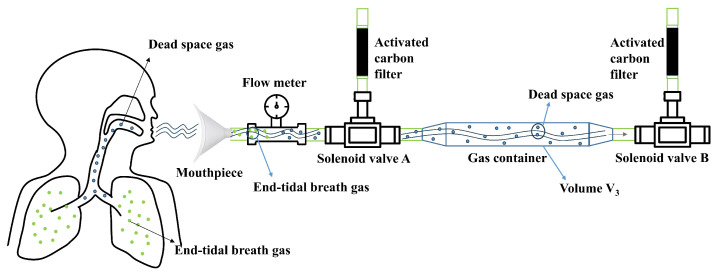
Gas distribution at time $t_{1}$.

**Figure 7 micromachines-16-00463-f007:**
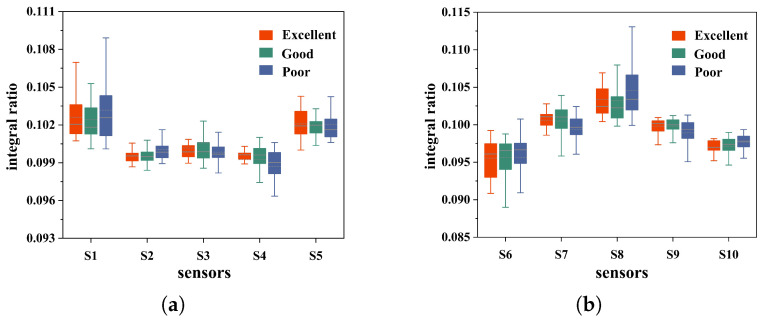
(**a**) Sensor 1–5 response data box plot. (**b**) Sensor 6–10 response data box plot.

**Figure 8 micromachines-16-00463-f008:**
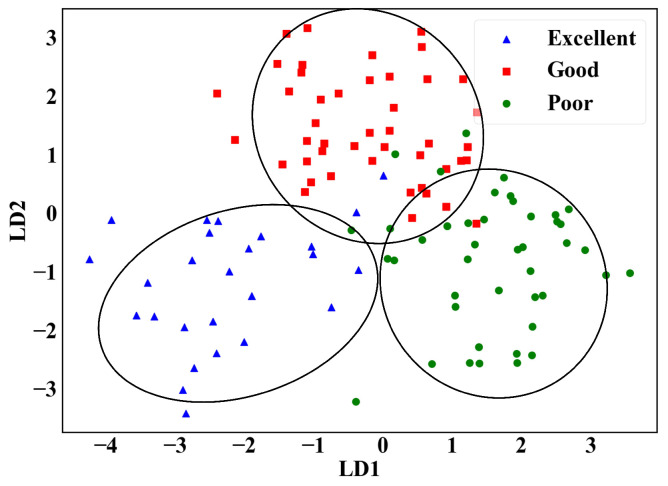
Results of LDA dimensionality reduction.

**Figure 9 micromachines-16-00463-f009:**
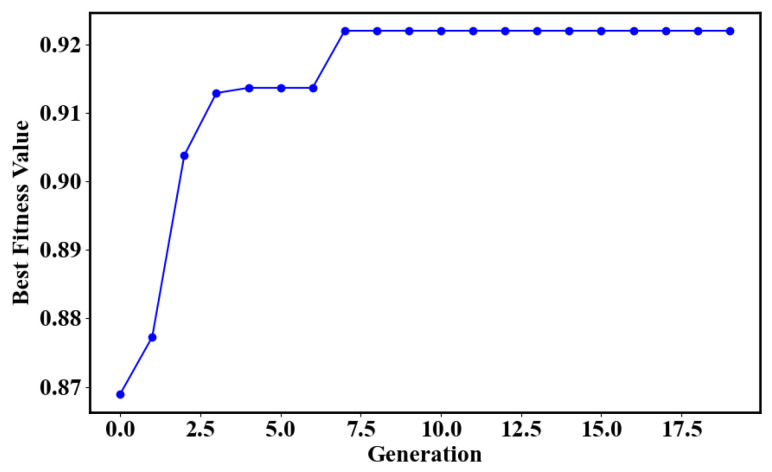
The graph illustrating the optimization of SVM parameters using GA.

**Table 1 micromachines-16-00463-t001:** Comparison of classification accuracy before and after optimization.

Parameter Combination (*c*,γ)	(0.01,1)	(0.1,1)	(1,1)	(1,10)	(10,100)	GA(1,0.2)
Accuracy	39.49%	62.29%	69.25%	91.19%	50.95%	92.06%

**Table 2 micromachines-16-00463-t002:** Classification results of three algorithms for different health conditions.

	Index	Excellent	Good	Poor
LDA-SVM	Accuracy		92.06%	
	Recall	92%	93.33%	90.83%
	Precision	96.67%	90.14%	93.06%
	F1 Score	93.74%	91.44%	91.88%
LDA-RF	Accuracy		90.32%	
	Recall	92%	91.11%	88.33%
	Precision	93.81%	90.14%	91.28%
	F1 Score	92.20%	90.12%	89.61%
LDA-KNN	Accuracy		93.04%	
	Recall	92.31%	97.78%	88.64%
	Precision	96%	88%	97.50%
	F1 Score	94.12%	92.63%	92.86%

## Data Availability

The sensor data will be made available upon request.

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
