# Peer review of "Design of an Electronic Nose System with Automatic End-Tidal Breath Gas Collection for Enhanced Breath Detection Performance"

_micromachines, 2025, doi:10.3390/mi16040463_

Round 1
Reviewer 1 Report
Comments and Suggestions for Authors
The manuscript presents an innovative electronic nose system capable of automatically collecting end-tidal breath gases and classifying human health status with high accuracy. The study introduces a threshold-based separation method for optimizing gas collection and integrates solenoid valves to ensure precise control over the gas path. The combination of structural improvements and automated control minimizes external interference and enhances the system’s performance. The experimental results demonstrate that the classification accuracy of SVM, RF, and KNN exceeds 90%, outperforming similar studies in breath-based health monitoring. The study aligns well with the scope of Micromachines and provides valuable insights into electronic nose technology for non-invasive health monitoring. However, there are several aspects that require clarification and improvement before acceptance.
Q1. In the manuscript, the authors state: "Studies have shown that adults exhale approximately 500 mL of gas per breath. The first 150 mL are dead space gases from the respiratory tract and nasopharynx, followed by 350 mL of end-tidal breath gases from the lungs [1]." However, this claim is supported by only a single reference, which is insufficient to establish a widely accepted parameter. The authors should include additional references to support this claim.
Q2. The manuscript describes a pre-experiment stage requiring three exhalations before formal sampling, but it does not address whether this process could affect biomarker concentrations. Continuous exhalation may lead to the depletion of certain breath biomarkers if they are not replenished quickly enough, potentially lowering their concentration in the formal collection stage. It is recommended to discuss this possible effect and ensure biomarker stability.
Q3 In Figure 3, the PWM duty cycle of vacuum pumps D and E is set to 50% during the cleaning state but reduced to 5% during the detection state. However, the manuscript does not explain the reason for this change. The authors should provide a clear rationale for adjusting the PWM duty cycle and discuss how it impacts the gas flow dynamics and the stability of the detection process.
Q4. The manuscript does not provide a clear justification for selecting the specific commercial sensors (MQ-138, MQ-136, MQ-8, MQ-5, MQ-7B, WSP2110, MQ-4B, MP-9, MP-4, MP-135) for the sensor array. The authors should explain the criteria for choosing these sensors and how their target gas sensitivities align with the study’s objectives. Additionally, some of these sensors may require high-temperature operation for optimal performance. It is recommended that the authors clarify whether these sensors operate under high-temperature conditions.
Q5. The manuscript does not specify the size and measurement range of the flow meter used in the system. To ensure clarity and reproducibility, the authors should provide details on the flow meter's capacity and accuracy.
Q6. The manuscript states that subjects' health conditions were classified into excellent (excellent), good (good), and poor (poor) based on their scores. However, it does not clearly explain how these health scores were calculated. The authors should provide a detailed explanation of the scoring criteria.
Q7. The manuscript states that LDA (Linear Discriminant Analysis) was used for dimensionality reduction before feeding the data into the classification model (Figure 8). However, it does not specify the final data dimensions after LDA reduction. Additionally, the study mentions using Integration Ratio, Maximum Location, First-Order Derivative’s Maximum, and Two Frequency Domain Eigenvalues as extracted features. It is unclear whether the data input to the classification model consists of the five extracted features or the LDA-reduced data. The authors should clarify the exact data representation used for classification and specify whether LDA-reduced data or the extracted feature set was ultimately fed into the classification model.
Q8. The manuscript states that a genetic algorithm (GA) was used for hyperparameter optimization, but Table 2 lists LDA-SVM, and the specific role of LDA in the optimization process is not clearly explained. The authors should clarify how LDA is applied in the model, whether it is used for feature selection or dimensionality reduction, and how it interacts with GA-based optimization. Additionally, the manuscript lacks a comparison of classification accuracy before and after genetic algorithm optimization. To validate the effectiveness of GA tuning, the authors should provide accuracy results for both cases to demonstrate the improvement achieved through hyperparameter optimization.
Reviewer 2 Report
Comments and Suggestions for Authors
The authors present an electronic nose system, which collects end-tidal breath from patients to predict the subject's health condition using LDA-SVM, LDA-RF and LDA-KNN classification algorithms. It turns out to be essential to probe only end-tidal breath samples and discarding the first portion of breath collected. The selection of end-tidal breath is achieved by a solenoid valve and flow meter system.
Some issues need to be clarified:
1) line 19: diverse array of volatile ... should read: diverse volatile organic compounds
2) Despite of listing the sensors in Table A1, it is not clear which type of sensors were used. I assume that they are metal oxide sensors, but please insert a paragraph in the text describing the sensors more precisely.
3) What are typical flow rates and pressures in the gas flow system? Are there any precautions taken to avoid adsorption of volatile compounds on surfaces in the flow system, in particular in solenoid valves?
Reviewer 3 Report
Comments and Suggestions for Authors
The manuscript entitled, ‘Design of an Electronic Nose System with Automatic End-Tidal Breath Gas Collection for Enhanced Breath Detection Performance’ reported Electronic Nose System with Automatic End-Tidal Breath Gas Collection for Enhanced Breath Detection. The article should be modified according the following comments:
- The abstract lacks specificity regarding the data presented in the study. It is recommended to highlight key findings or notable data points to give readers a clearer understanding of the study's contributions.
- How do temperature and humidity specifically affect the sensitivity of the gas sensors, and can this is quantified?
- Could you provide more details or justification for choosing conductivity ratio over raw voltage as the preferred measurement method?
- What were the exact criteria for selecting the classification performance metrics (accuracy, recall, precision, F1 score)? Were any other metrics considered?
- Can you explain the rationale behind using LDA for dimensionality reduction before applying the classification algorithms, and were any alternative methods tested?
- What challenges were encountered in applying traditional machine learning algorithms to the data, and how were they overcome?
- How did the improved data collection method influence the classification model’s performance, and can the approach be generalized to other applications or datasets?
- How was the quality and representativeness of the experimental data ensured during the data collection process, and were there any challenges in this regard?
- What role did the exhaled gas flow and gas separation threshold play in the accuracy of the data collection, and how were these parameters determined?
- Can you clarify how the classification performance of this study compares with other similar studies, and why your results outperform the existing literature?
- Some articles would be significance for your reference:
- Das, P., Marvi, P. K., Ganguly, S., Tang, X., Wang, B., Srinivasan, S., ... & Rosenkranz, A. (2024). MXene-based elastomer mimetic stretchable sensors: design, properties, and applications. Nano-Micro Letters, 16(1), 135.
- Ward, A., Petrie, A., Honek, J. F., & Tang, X. S. (2014). Analyte-Dependent Sensing Mechanisms: The Fabrication and Characterization of a 32-Channel Array of SWCNT-TF Chemiresistive Sensors. IEEE Nanotechnology Magazine, 8(2), 29-37.
Round 2
Reviewer 3 Report
Comments and Suggestions for Authors
This can be published in its present form.